# Suspicious Ultrasound-Occult Non-Calcified Mammographic Masses, Asymmetries, and Architectural Distortions Are Moderate Probability for Malignancy

**DOI:** 10.3390/cancers16030655

**Published:** 2024-02-03

**Authors:** Ethan O. Cohen, Rachel E. Perry, Ravinder S. Legha, Hilda H. Tso, Kyungmin Shin, Megan E. Speer, Kanchan A. Phalak, Jia Sun, Jessica W. T. Leung

**Affiliations:** 1Department of Breast Imaging, Division of Diagnostic Imaging, The University of Texas MD Anderson Cancer Center, Houston, TX 77030, USA; reperry1@mdanderson.org (R.E.P.); rlegha@mdanderson.org (R.S.L.); htso@mdanderson.org (H.H.T.); kshin1@mdanderson.org (K.S.); mspeer@mdanderson.org (M.E.S.); kphalak@mdanderson.org (K.A.P.); jwleung@mdanderson.org (J.W.T.L.); 2Department of Biostatistics, The University of Texas MD Anderson Cancer Center, Houston, TX 77030, USA; jsun9@mdanderson.org

**Keywords:** mammography, breast cancer, ultrasound, tomosynthesis, architectural distortion, breast biopsy

## Abstract

**Simple Summary:**

Non-calcified mammographic findings, which are suspicious for breast cancer, often undergo ultrasound for further evaluation and to facilitate percutaneous tissue sampling. Most of these findings have a correlation with ultrasound examinations, but some do not. The significance of those without an ultrasound correlation has not been evaluated with modern mammographic techniques. Specifically, mammographic architectural distortion is of particular interest given that it is more commonly identified with digital breast tomosynthesis, a modern quasi-three-dimensional mammographic technique. This article serves to provide management guidance for those situations by retrospectively evaluating a very large cohort of these findings.

**Abstract:**

Suspicious non-calcified mammographic findings have not been evaluated with modern mammographic technique, and the purpose of this work is to compare the likelihood of malignancy for those findings. To do this, 5018 consecutive mammographically guided biopsies performed during 2016–2019 at a large metropolitan, community-based hospital system were retrospectively reviewed. In total, 4396 were excluded for targeting calcifications, insufficient follow-up, or missing data. Thirty-seven of 126 masses (29.4%) were malignant, 44 of 194 asymmetries (22.7%) were malignant, and 77 of 302 architectural distortions (AD, 25.5%) were malignant. The combined likelihood of malignancy was 25.4%. Older age was associated with a higher likelihood of malignancy for each imaging finding type (all *p* ≤ 0.006), and a possible ultrasound correlation was associated with a higher likelihood of malignancy when all findings were considered together (*p* = 0.012). Two-view asymmetries were more frequently malignant than one-view asymmetries (*p* = 0.03). There were two false-negative biopsies (98.7% sensitivity and 100% specificity). In conclusion, the 25.4% likelihood of malignancy confirms the recommendation for biopsy of suspicious, ultrasound-occult, mammographic findings. Mammographically guided biopsies were highly sensitive and specific in this study. Older patient age and a possible ultrasound correlation should raise concern given the increased likelihood of malignancy in those scenarios.

## 1. Introduction

Breast cancer is now the leading cause of new cancer diagnoses in women in the United States, and incidence rates have risen by approximately 0.5% per year since the early 2000s [1]. The American woman’s lifetime risk of invasive breast cancer is approximately 13%, but thankfully, mortality has decreased by 42% since peaking in 1989 due to increased awareness, more effective treatment, and improved mammographic screening, the latter of which has been shown to reduce breast cancer-related deaths by up to 40–45% [1,2,3,4].

The fifth edition of the American College of Radiology’s BI-RADS Atlas describes four common mammographic presentations for breast cancer: calcifications, masses, architectural distortion (AD), and asymmetries [5]. Management algorithms for calcifications are well established, and ultrasounds are often not performed [6,7]. However, ultrasounds are included in the routine evaluation of the remaining findings because it allows for further lesion characterization and can be used for biopsy guidance when indicated [5]. Prior studies have focused on the likelihood of malignancy for these mammographic findings. In 2009, Venkatesan et al. prospectively evaluated 10,641 mammograms and calculated a positive predictive value of 3.6–12.7% for screening recall of masses, calcifications, asymmetries, and ADs and a positive predictive value of 14.6–60.2% for masses, calcifications, asymmetries, and ADs identified at diagnostic mammography [8]. More recently Hu et al. noted a likelihood of malignancy of 4.5–28.2% for the same mammographic findings recalled from screening [9], while older data has demonstrated a 0.7–12.8% likelihood of malignancy for the four types of mammographic asymmetries (asymmetry, focal asymmetry, global asymmetry, and developing asymmetry) [10,11].

The increased adoption of digital breast tomosynthesis (DBT) has led to more studies on AD, which is increasingly apparent with DBT and has a high positive predictive value for malignancy [8,9,12,13]. Among those prior studies, many have compared AD with a correlation on ultrasounds to AD without. In general, suspicious AD with an ultrasound correlate has a high level of suspicion for malignancy (BI-RADS assessment category 4c), whereas suspicious AD without an ultrasound correlate has a moderate level of suspicion for malignancy (BI-RADS assessment category 4b) [5,13,14,15,16,17,18,19,20]. However, the studies evaluating AD without an ultrasound correlation are of limited sample size, and no prior study has also included ultrasound-occult masses or asymmetries. As such, the purpose of our study was to calculate and compare the likelihoods of malignancy for suspicious, ultrasound-occult masses, asymmetries, and ADs. We hypothesized that the likelihood of malignancy for masses and ADs in our study would be significantly higher than asymmetries and that suspicious ultrasound-occult asymmetries might even approach the 0–2% likelihood of malignancy for probably benign BI-RADS 3 findings.

## 2. Materials and Methods

This Health Insurance Portability and Accountability Act-compliant, retrospective study was approved by the institutional review board. Written informed consent was waived. We retrospectively reviewed all mammographically guided breast biopsies from a large, metropolitan community-based health system performed between 1 January 2016 and 31 December 2019. The procedures occurred at one of nine community-based hospitals or their affiliated imaging centers. Note that the study populations from two prior publications were included in their entirety within this study’s population. The first evaluated how the needle approach for prone DBT-guided biopsy affected hematoma formation and clip migration [21], and the second directly compared procedural details for prone and upright, stereotactic, and DBT-guided breast biopsies [19]. The study population from a third prior publication evaluating DBT screening in women with breast implants partially overlapped with this current study population [22].

The following procedures were excluded: biopsies targeting pure calcifications; biopsies targeting calcifications associated with a mass, asymmetry, or AD; procedures yielding benign results without at least one year of benign imaging follow-up; and cases with incomplete data within the medical record. As such, ground truth was defined as malignant pathology or benign pathology with at least one year of negative or benign imaging follow-up (Figure 1).

During the study time period, all diagnostic mammograms were reviewed in real-time by a radiologist to determine if further mammographic views or ultrasound were necessary for complete evaluation. Masses demonstrating growth or suspicious shapes or margins, new or increasing persistent asymmetries, and ADs without a characteristically benign etiology were considered suspicious unless ultrasound revealed a characteristically benign etiology. Mammographically guided biopsy was recommended for these findings when an ultrasound failed to reveal a definite correlation. Further imaging details have been previously described [22].

Mammographically guided breast biopsies were performed by one of 39 radiologists with 1–41 years of experience (mixture of academic, private-practice, fellowship-trained, and non-fellowship-trained physicians). Either DBT or stereotaxis with digital mammography (DM) was used for imaging guidance during tissue sampling (as recommended by the radiologist interpreting the diagnostic mammogram and/or as determined by the radiologist performing the biopsy procedure). DBT-guided breast biopsy was not available at all facilities throughout the entire study period, but it was available at most, so patients in need of a DBT-guided biopsy were scheduled at a facility with that capability. The biopsy procedures and equipment have also been previously described in detail [19,21].

Patient-level data retrospectively collected from the medical record (Centricity, GE Healthcare; Millennium, Cerner) included age, mammographic breast density, personal history of breast cancer, and family history of breast cancer. Race and ethnicity were unknown or erroneously documented for most patients in the study. The following data were recorded for each suspicious mammographic lesion: initial mammography examination revealing the lesion (DM or DBT), indication for diagnostic mammography, lesion type (mass, asymmetry, or AD based on the diagnostic mammogram report), lesion size on the diagnostic mammogram, visualization of the mammographic lesion on one or two orthogonal mammographic projections, and the presence of a possible, though not definitive, ultrasound correlation for the mammographic finding. For purposes of our study, a definite ultrasound correlation was defined as a corresponding lesion on ultrasound that could be adequately and confidently biopsied percutaneously using ultrasound guidance. This was determined by the radiologist evaluating the diagnostic mammogram and ultrasound, and it was recorded within the ultrasound report. Of note, there was overlap in lesion types in the mammography reports (e.g., “mass with architectural distortion”), but our electronic record afforded us the ability to determine the principal lesion type, as determined by the interpreting radiologist, which was used for analysis. Associated findings were not recorded.

Outcomes were also recorded for each procedure, including radiologic-pathology concordance, percutaneous biopsy histopathology, surgical histopathology, and imaging follow-up results. When procedures yielded more than one high-risk pathology, the following rank order was used to determine the primary high-risk result (from first to last): atypical ductal hyperplasia, lobular neoplasia (classic lobular carcinoma in situ or atypical lobular hyperplasia), radial scar or complex sclerosing lesion (hereafter collectively referred to as radial sclerosing lesions), papilloma, flat epithelial atypia, and atypia not otherwise specified. Of note, there were no cases of florid or pleomorphic lobular carcinoma in situ.

Patient demographics, clinical data, and imaging characteristics were summarized using frequencies, percentages, means, standard deviations, medians, minimums, and maximums. They were subsequently correlated with lesion type using the Wilcoxon rank sum test or Fisher’s exact test. Positive predictive values (PPVs) were compared among lesion types using Pearson’s Chi-squared test. Patient age, prior history of breast cancer, lesion size, etc. were correlated with malignant outcomes using the Wilcoxon rank sum test or Fisher’s exact test. *p*-value less than 0.05 was considered statistically significant. Statistical analyses were carried out using R (version 3.6.3, R Development Core Team, Vienna, Austria). 

## 3. Results

### 3.1. Study Population

After exclusions, the final study population consisted of 607 patients (mean 57 ± 11 years old) with 622 suspicious, non-calcified, mammographic lesions that underwent mammographically guided biopsy (Figure 1). Nearly half of the biopsy targets were ADs (302 of 622, 48.6%), while the remainder were masses (126 of 622, 20.2%) and asymmetries (194 of 622, 31.2%). Ultimately, the likelihoods of malignancy for suspicious masses, asymmetries, and ADs were similar: 29% (37/126), 23% (44/194), and 25% (77/302), respectively (*p* = 0.40).

Table 1 compares patient demographics and lesion characteristics between lesion types, and several important differences deserve mention here. Specifically, mammographic breast density, a known breast cancer risk factor [23], was significantly higher in cases of AD compared with masses and asymmetries. Also, masses included in the study were smaller, more commonly seen on two mammographic views, and more often associated with a possible ultrasound correlate than AD or asymmetries.

### 3.2. Benign Versus Malignant Outcomes

Table 2 compares patient demographics and lesion characteristics based on benign and malignant outcomes. Certain important results deserve specific mention here. First, the likelihoods of malignancy for suspicious masses, asymmetries, and Ads were similar: 29% (37/126), 23% (44/194), and 25% (77/302), respectively (*p* = 0.40). Increased age was associated with malignancy for each imaging finding type (*p* ≤ 0.006), and a possible ultrasound correlate was associated with a higher likelihood of malignancy when all findings were considered together (*p* = 0.012). Two-view asymmetries were more commonly malignant than one-view asymmetries (*p* = 0.03). Finally, there were two false-negative biopsy results in our study. The first was a suspicious asymmetry biopsied in a 59-year-old female. Biopsy pathology was sclerosing adenosis, which was deemed concordant, but subsequent diagnostic imaging later revealed an invasive ductal carcinoma. The second false-negative biopsy result occurred in a 66-year-old female with architectural distortion that yielded benign fat necrosis at mammographically guided breast biopsy (Figure 2, twelve 9-gauge samples acquired). This result was classified as concordant, but diagnostic imaging less than one year later revealed a mass at the site of the architectural distortion that was found to represent a grade two invasive ductal carcinoma. Calculated sensitivity and specificity are thus, 98.7% and 100%, respectively.

Table 3 compares the likelihood of malignancy for each lesion type based on its initial visualization at screening mammography or diagnostic mammography, indicating no statistically significant difference between suspicious lesions seen at screening or diagnostic imaging. Table 4 similarly compares outcomes based on whether the suspicious lesion was initially identified on DM or DBT, and again, there was no significant difference in the likelihood of malignancy of suspicious lesions identified at DM or DBT. However, suspicious, ultrasound-occult AD was significantly more likely to be identified on DBT, as opposed to DM, than either masses or asymmetries (*p* < 0.001).

### 3.3. Architectural Distortions

Figure 3 illustrates the final outcomes for all suspicious, ultrasound-occult AD. Again, 25% were found to represent breast malignancy. Of note, 105 of the 221 benign concordant histologies were radial sclerosing lesions (77 excised and 28 observed), and only two upgraded to carcinoma (1.9%, one invasive lobular carcinoma and one ductal carcinoma in situ). The median number of 9-gauge vacuum-assisted biopsy passes made during each of these 105 procedures was ten (range 4–19; mean 9.9 ± 2.6), and the two cases that upgraded to carcinoma were sampled with four and twelve biopsy passes.

## 4. Discussion

Our retrospective review of over 5000 mammographically guided breast biopsies yielded many important results. First, the likelihoods of malignancy for suspicious, ultrasound-occult masses (29.4%), asymmetries (22.7%), and AD (25.5%) were all within the range of the moderate probability BI-RADS assessment category 4b, i.e., between 10 and 50% [5]. The sensitivity and specificity of mammographically guided biopsy for these lesions was extremely high, 98.7% and 100%, respectively. Older age was associated with a higher likelihood of malignancy for each imaging finding type, while a possible ultrasound correlation was associated with a higher likelihood of malignancy when all findings were considered together. Two-view asymmetries were more frequently malignant than one-view asymmetries, and lesion size did not correlate with likelihood of malignancy. These findings should help clarify management recommendations for suspicious, ultrasound-occult, mammographic lesions, in particular AD, which is increasingly identified with DBT [9,12,13].

Overall, there is a relative paucity of recent literature investigating the likelihood of malignancy for each mammographic lesion type, particularly in the community imaging setting as our study was performed in. Venkatesan et al. prospectively evaluated screening and diagnostic mammography findings in 10,262 women between 1998 and 2002 and found the following positive predictive values for the four common mammographic lesion types: masses (19.6%), calcifications (24.1%), asymmetries (14.6%), and AD (60.2%) [8]. Similar to our study, they noted that the likelihood of malignancy increased with age. However, their study was dissimilar to ours in that all lesions included in our study were evaluated with diagnostic sonography and were determined to have no definite corresponding ultrasound finding. A 2019 report by Hu et al. [9] found positive predictive values at biopsy similar to the findings from Venkatesan et al. (36.9% for masses, 26.5% for calcifications, 26.6% for focal asymmetries, and 58.9% for AD), though their study focused only on mammographic findings recalled from screening and again did not delineate between findings with and without an ultrasound correlation. Other data focusing on asymmetries agree, with reported likelihoods of malignancy varying between 0.67% and 26.7% depending on the asymmetry type and whether it was identified initially at screening or diagnostic mammography [8,9,10,11,24,25]. Overall, the likelihoods of malignancy for ultrasound-occult asymmetries and masses in our study are similar to these prior reports, while the likelihood of malignancy for ultrasound-occult AD in our study was lower, assumedly because we included only cases without a definite ultrasound correlate.

This sub-analysis of 302 cases of ultrasound-occult AD represents the largest sample size reported to date, and the calculated likelihood of malignancy in this retrospective review was 25.5%. Again, this is lower than the studies mentioned above which did not distinguish between AD visible at ultrasound and AD occult at ultrasound, yet it is very much in line with more recent reports that have found a 13–29% likelihood of malignancy for AD without a definite ultrasound correlate [14,15,16,17,18,19,20,26,27]. These, and our data confirm the moderate probability BI-RADS assessment category 4b for suspicious, ultrasound-occult AD. Further work has compared AD identified at DM with AD identified at DBT, and these studies demonstrated higher likelihoods of malignancy for AD identified at DM versus DBT [26,27]. We performed a similar comparison in our study, but we found no significant difference in the likelihoods of malignancy between ultrasound-occult AD detected at DM and ultrasound-occult AD detected at DBT (*p* = 0.61). This is possibly due to our small sample size, despite our sub-analysis of ultrasound-occult AD being the largest of its kind, or it might represent an important conclusion. Further work here is indicated. Finally, we found a significantly higher percentage of suspicious ultrasound-occult AD identified at DBT rather than DM, compared with masses and asymmetries (*p* < 0.001). This trend is not unexpected given that DBT is known to reveal more AD than DM [9,12,13].

Comparison of patient and imaging characteristics for suspicious, ultrasound-occult, mammographic lesions yielded further interesting findings. First, increased patient age was significantly associated with malignant outcomes for all lesion types, which is not surprising given prior data and the fact that age is a very important breast cancer risk factor [8,28,29,30]. Our findings that both a possible ultrasound correlation and visualization of an asymmetry on two mammographic views, as opposed to one, correlate with an increased likelihood of malignancy are also concordant with existing literature [8,11,13,20,25,26,27,31,32]. Interestingly, we did not identify a higher likelihood of malignancy for lesions identified at diagnostic mammography compared with those identified at screening mammography as in the 2009 Venkatesan et al. study [8]. This could also be due to our small sample size, but it is certainly unexpected since suspicious findings at diagnostic mammography have a higher pre-test probability than those at screening mammography [5]. Finally, important note should be made of the outcomes for the 105 radial sclerosing lesions in our study. Only two of these cases were associated with malignancy, yielding an upgrade rate of 1.9%, which is in-line with prior data revealing a 1% upgrade rate for radial sclerosing lesions without atypia sampled with vacuum-assistance and an 8–11 gauge needle [33]. Imaging follow-up in lieu of surgical excision might be appropriate here.

Our study had several limitations. First, our study design included only suspicious, non-calcified, ultrasound-occult lesions that underwent mammographically guided biopsy. Benign lesions, partially calcified lesions, and cases evaluated with short-interval follow-up imaging instead of biopsy were not included in our dataset, and our study did not evaluate mammographic findings with an ultrasound correlate as an internal comparison. Including these extra cases would have yielded additional important conclusions about the significance of each mammographic finding type. We also had quite a few cases that were lost to follow-up and did not undergo surgery or at least 12 months of imaging follow-up. Moreover, we were unable to elucidate whether the suspicious mammographic lesions were initially seen, only seen, or even better seen on the synthetic mammogram, DM, or DBT views, which might have led to additional conclusions regarding the significance of each finding type. Finally, our retrospective study design limits generalizability, though we believe that including a mixture of academic, private-practice, fellowship-trained, and non-fellowship-trained radiologists working within a large community-based healthcare system helps to mitigate this limitation.

## 5. Conclusions

In conclusion, the overall likelihood of malignancy for suspicious, non-calcified mammographic findings in this review of over 5000 mammographically guided breast biopsies was 25.4%, confirming a BI-RADS category assessment of moderate suspicion for malignancy, BI-RADS 4b. Mammographically guided biopsy of these lesions was extremely sensitive and specific. Increased age and a possible ultrasound correlation should raise concern due to the increased likelihood of malignancy in these scenarios.

## Figures and Tables

**Figure 1 cancers-16-00655-f001:**
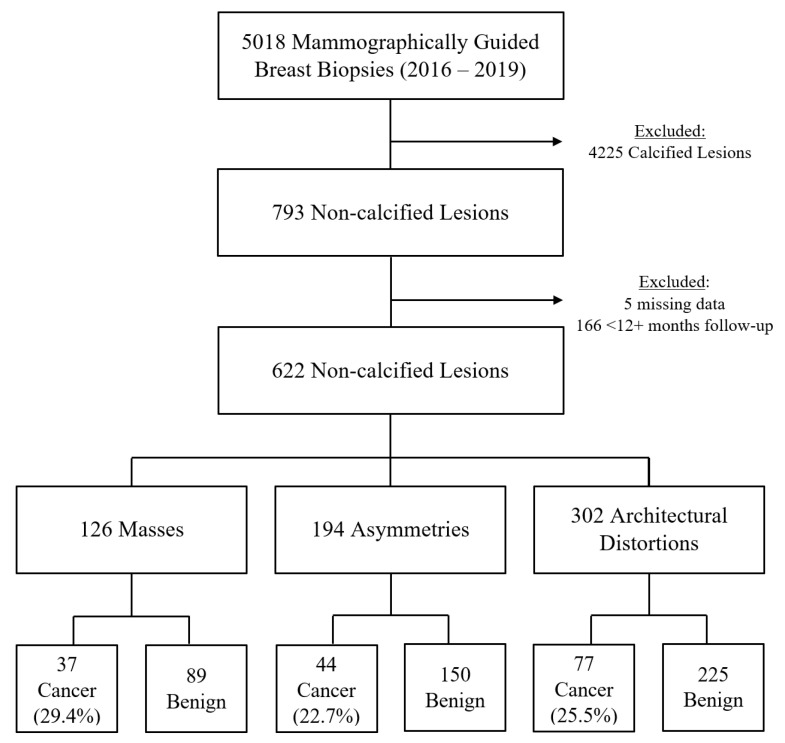
Flowchart illustrating the retrospective study design, biopsy outcomes, and likelihoods of malignancy.

**Figure 2 cancers-16-00655-f002:**
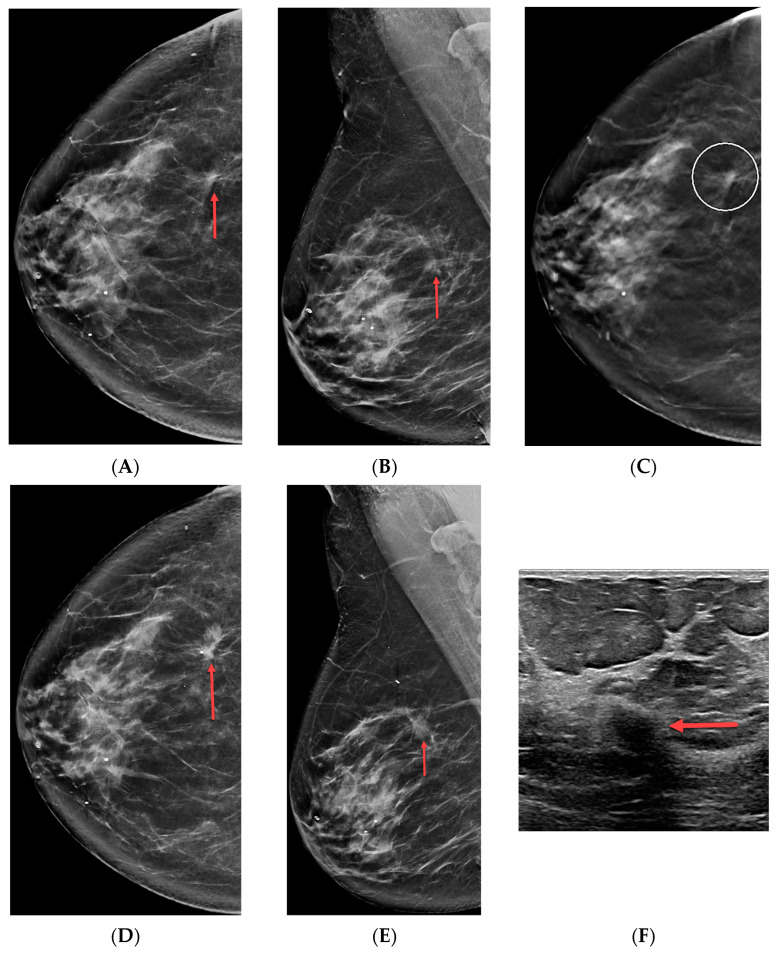
A 66-year-old female with ultrasound-occult architectural distortion. (**A**) Right craniocaudal synthetic mammogram, (**B**) right mediolateral oblique synthetic mammogram, and (**C**) right craniocaudal tomosynthesis slice from 2018 demonstrate suspicious architectural distortion (red arrows and open circle). An ultrasound correlate was not identified, and benign fat necrosis was diagnosed at tomosynthesis-guided biopsy. This result was deemed concordant. The patient’s (**D**) right craniocaudal synthetic mammogram, (**E**) right mediolateral oblique synthetic mammogram, and (**F**) targeted ultrasound from 2019 reveal a grade two invasive ductal carcinoma (red arrows) at the site of mammographic concern from 2018, indicating that fat necrosis was a false-negative biopsy result.

**Figure 3 cancers-16-00655-f003:**
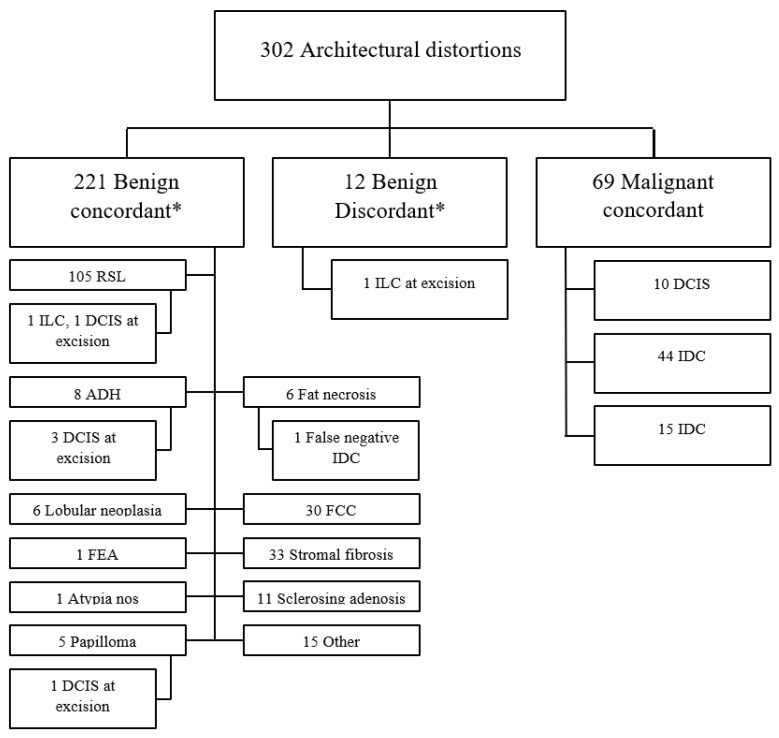
Flowchart demonstrating outcomes for suspicious ultrasound-occult architectural distortions. * Ground truth is based on either excision or at least twelve months of benign imaging follow-up. ADH, atypical ductal hyperplasia; DCIS, ductal carcinoma in situ; FCC, fibrocystic changes; FEA, flat epithelial atypia; IDC, invasive ductal carcinoma; ILC, invasive lobular carcinoma; RSL, radial sclerosing lesion.

**Table 1 cancers-16-00655-t001:** Patient and imaging characteristics by biopsy target lesion type.

	Masses (N = 126)	Asymmetries (N = 194)	Distortions (N = 302)	*p* Value
Age (years)				0.989
Mean (SD)	56.99 (10.75)	57.12 (11.70)	56.89 (10.84)	
Median (Range)	57.76 (35.89, 80.97)	56.18 (36.52, 86.20)	56.30 (34.20, 88.00)	
Breast density				**<0.001**
A	11 (8.73%)	15 (7.73%)	3 (0.99%)	
B	72 (57.14%)	104 (53.61%)	82 (27.15%)	
C	43 (34.13%)	71 (36.60%)	197 (65.23%)	
D	0 (0%)	4 (2.06%)	20 (6.62%)	
Personal history of breast cancer				0.503
No	121 (96.03%)	180 (92.78%)	284 (94.04%)	
Yes	5 (3.97%)	14 (7.22%)	18 (5.96%)	
Family history of breast cancer				0.057
No	72 (57.14%)	119 (61.34%)	207 (68.54%)	
Yes	54 (42.86%)	75 (38.66%)	95 (31.46%)	
Recently diagnosed with breast cancer ^†^				1
No	124 (98.41%)	191 (98.45%)	298 (98.68%)	
Yes	2 (1.59%)	3 (1.55%)	4 (1.32%)	
Lesion size (millimeter)				**<0.001**
N *	115	165	256	
Mean (SD)	9.53 (6.42)	19.11 (19.01)	16.19 (10.16)	
Median (Range)	8.00 (3.00, 46.00)	13.00 (3.00, 132.00)	14.00 (2.00, 63.00)	
Seen on 1 or 2 mammographic views				**<0.001**
1	2 (1.59%)	29 (14.95%)	30 (9.93%)	
2	124 (98.41%)	165 (85.05%)	272 (90.07%)	
Possible ultrasound correlate				**<0.001**
No	74 (58.73%)	158 (81.44%)	204 (67.55%)	
Yes	52 (41.27%)	36 (18.56%)	98 (32.45%)	
Reason for diagnostic mammogram				0.087
Screening recall	104 (82.54%)	151 (77.84%)	246 (81.46%)	
BI-RADS 3 follow-up	5 (3.97%)	13 (6.70%)	3 (0.99%)	
Palpable lump	8 (6.35%)	10 (5.15%)	20 (6.62%)	
Prior history of breast cancer	2 (1.59%)	5 (2.58%)	8 (2.65%)	
Other	7 (5.56%)	15 (7.73%)	25 (8.28%)	

Note—Data are the number (%) of target lesions unless otherwise stated. Boldface indicates statistical significance (*p* ≤ 0.05). SD, standard deviation. * Cases with unknown lesion size were not included herein. ^†^ Recently is defined herein as within the three months preceding the diagnostic mammogram. Boldface indicates statistical significance (*p* ≤ 0.05).

**Table 2 cancers-16-00655-t002:** Biopsy outcomes by imaging or patient characteristics per biopsy target lesion type.

	Masses				Asymmetry				Architectural Distortion				Combined			
	Benign (N = 89)	Cancer (N = 37)	Total (N = 126)	*p* Value	Benign (N = 150)	Cancer (N = 44)	Total (N = 194)	*p* Value	Benign (N = 225)	Cancer (N = 77)	Total (N = 302)	*p* Value	Benign (N = 464)	Cancer (N = 158)	Total (N = 622)	*p* Value
Age (years)				**<0.001**				**<0.001**				**0.006**				**<0.001**
Mean (SD)	54.60 (10.38)	62.74 (9.47)	56.99 (10.75)		54.43 (10.68)	66.26 (10.42)	57.12 (11.70)		55.92 (11.04)	59.72 (9.76)	56.89 (10.84)		55.18 (10.80)	62.25 (10.20)	56.98 (11.08)	
Median (Range)	53.02 (35.89, 80.97)	63.90 (40.06, 76.07)	57.76 (35.89, 80.97)		51.88 (36.52, 80.81)	66.22 (44.96, 86.20)	56.18 (36.52, 86.20)		54.72 (34.20, 88.00)	59.48 (41.00, 80.63)	56.30 (34.20, 88.00)		53.47 (34.20, 88.00)	62.63 (40.06, 86.20)	56.41 (34.20, 88.00)	
Breast density				0.55				0.28				0.67				0.86
A	9 (10.11%)	2 (5.41%)	11 (8.73%)		9 (6.00%)	6 (13.64%)	15 (7.73%)		2 (0.89%)	1 (1.30%)	3 (0.99%)		20 (4.31%)	9 (5.70%)	29 (4.66%)	
B	48 (53.93%)	24 (64.86%)	72 (57.14%)		80 (53.33%)	24 (54.55%)	104 (53.61%)		64 (28.44%)	18 (23.38%)	82 (27.15%)		192 (41.38%)	66 (41.77%)	258 (41.48%)	
C	32 (35.96%)	11 (29.73%)	43 (34.13%)		58 (38.67%)	13 (29.55%)	71 (36.60%)		143 (63.56%)	54 (70.13%)	197 (65.23%)		233 (50.22%)	78 (49.37%)	311 (50.00%)	
D	0 (0%)	0 (0%)	0 (0%)		3 (2.00%)	1 (2.27%)	4 (2.06%)		16 (7.11%)	4 (5.19%)	20 (6.62%)		19 (4.09%)	5 (3.16%)	24 (3.86%)	
Personal history of breast cancer				0.150				1.0				0.58				1
No	87 (97.75%)	34 (91.89%)	121 (96.03%)		139 (92.67%)	41 (93.18%)	180 (92.78%)		210 (93.33%)	74 (96.10%)	284 (94.04%)		436 (93.97%)	149 (94.30%)	585 (94.05%)	
Yes	2 (2.25%)	3 (8.11%)	5 (3.97%)		11 (7.33%)	3 (6.82%)	14 (7.22%)		15 (6.67%)	3 (3.90%)	18 (5.96%)		28 (6.03%)	9 (5.70%)	37 (5.95%)	
Family history of breast cancer				0.70				0.73				0.20				0.180
No	52 (58.43%)	20 (54.05%)	72 (57.14%)		93 (62.00%)	26 (59.09%)	119 (61.34%)		159 (70.67%)	48 (62.34%)	207 (68.54%)		304 (65.52%)	94 (59.49%)	398 (63.99%)	
Yes	37 (41.57%)	17 (45.95%)	54 (42.86%)		57 (38.00%)	18 (40.91%)	75 (38.66%)		66 (29.33%)	29 (37.66%)	95 (31.46%)		160 (34.48%)	64 (40.51%)	224 (36.01%)	
Recently diagnosed with breast cancer ^†^				0.50				0.129				1.0				0.24
No	88 (98.88%)	36 (97.30%)	124 (98.41%)		149 (99.33%)	42 (95.45%)	191 (98.45%)		222 (98.67%)	76 (98.70%)	298 (98.68%)		459 (98.92%)	154 (97.47%)	613 (98.55%)	
Yes	1 (1.12%)	1 (2.70%)	2 (1.59%)		1 (0.67%)	2 (4.55%)	3 (1.55%)		3 (1.33%)	1 (1.30%)	4 (1.32%)		5 (1.08%)	4 (2.53%)	9 (1.45%)	
Lesion size (millimeter)				0.88				0.44				0.32				0.117
N *	78	37	115		126	39	165		190	66	256		394	142	536	
Mean (SD)	9.32 (5.71)	9.97 (7.79)	9.53 (6.42)		18.19 (15.13)	22.08 (28.19)	19.11 (19.01)		16.70 (10.66)	14.71 (8.48)	16.19 (10.16)		15.72 (12.01)	15.50 (16.83)	15.66 (13.44)	
Median (Range)	8.00 (3.00, 37.00)	7.00 (3.00, 46.00)	8.00 (3.00, 46.00)		13.50 (3.00, 100.00)	10.00 (4.00, 132.00)	13.00 (3.00, 132.00)		15.00 (2.00, 63.00)	14.00 (2.00, 54.00)	14.00 (2.00, 63.00)		12.00 (2.00, 100.00)	10.50 (2.00, 132.00)	12.00 (2.00, 132.00)	
Seen on 1 or 2 mammographic views				0.085				**0.03**				1.0				0.22
1	0 (0.00%)	2 (5.41%)	2 (1.59%)		27 (18.00%)	2 (4.55%)	29 (14.95%)		23 (10.22%)	7 (9.09%)	30 (9.93%)		50 (10.78%)	11 (6.96%)	61 (9.81%)	
2	89 (100.00%)	35 (94.59%)	124 (98.41%)		123 (82.00%)	42 (95.45%)	165 (85.05%)		202 (89.78%)	70 (90.91%)	272 (90.07%)		414 (89.22%)	147 (93.04%)	561 (90.19%)	
Possible ultrasound correlate				0.33				0.121				0.162				**0.012**
No	55 (61.80%)	19 (51.35%)	74 (58.73%)		126 (84.00%)	32 (72.73%)	158 (81.44%)		157 (69.78%)	47 (61.04%)	204 (67.55%)		338 (72.84%)	98 (62.03%)	436 (70.10%)	
Yes	34 (38.20%)	18 (48.65%)	52 (41.27%)		24 (16.00%)	12 (27.27%)	36 (18.56%)		68 (30.22%)	30 (38.96%)	98 (32.45%)		126 (27.16%)	60 (37.97%)	186 (29.90%)	
Reason for diagnostic mammogram				0.74				0.151				0.49				0.188
Screening recall	73 (82.02%)	31 (83.78%)	104 (82.54%)		120 (80.00%)	31 (70.45%)	151 (77.84%)		183 (81.33%)	63 (81.82%)	246 (81.46%)		376 (81.03%)	125 (79.11%)	501 (80.55%)	
BI-RADS 3 follow-up	3 (3.37%)	2 (5.41%)	5 (3.97%)		10 (6.67%)	3 (6.82%)	13 (6.70%)		2 (0.89%)	1 (1.30%)	3 (0.99%)		15 (3.23%)	6 (3.80%)	21 (3.38%)	
Palpable lump	7 (7.87%)	1 (2.70%)	8 (6.35%)		7 (4.67%)	3 (6.82%)	10 (5.15%)		17 (7.56%)	3 (3.90%)	20 (6.62%)		31 (6.68%)	7 (4.43%)	38 (6.11%)	
Prior history of breast cancer	1 (1.12%)	1 (2.70%)	2 (1.59%)		5 (3.33%)	0 (0.00%)	5 (2.58%)		7 (3.11%)	1 (1.30%)	8 (2.65%)		13 (2.80%)	2 (1.27%)	15 (2.41%)	
Other	5 (5.62%)	2 (5.41%)	7 (5.56%)		8 (5.33%)	7 (15.91%)	15 (7.73%)		16 (7.11%)	9 (11.69%)	25 (8.28%)		29 (6.25%)	18 (11.39%)	47 (7.56%)	

Note—Data are number (%) of target lesions unless otherwise stated. Boldface indicates statistical significance (*p* ≤ 0.05). SD, standard deviation. * Cases with unknown lesion size were not included herein. **^†^** Recently is herein defined as within the three months preceding the diagnostic mammogram. Boldface indicates statistical significance (*p* ≤ 0.05).

**Table 3 cancers-16-00655-t003:** Biopsy outcomes based on initial visualization of each target lesion on screening or diagnostic mammography.

		Screening Mammogram	Diagnostic Mammogram	*p* Value
Mass	Benign (N = 89)	73 (82.02%)	16 (17.98%)	1.0
	Cancer (N = 37)	31 (83.78%)	6 (17.22%)	
Asymmetry	Benign (N = 150)	120 (80.00%)	30 (20.00%)	0.22
	Cancer (N = 44)	31 (70.45%)	13 (29.55%)	
Architectural Distortion	Benign (N = 225)	183 (81.33%)	42 (18.67%)	1.0
	Cancer (N = 77)	63 (81.82%)	14 (18.18%)	
Combined	Benign (N = 464)	376 (81.03%)	88 (18.97%)	0.64
	Cancer (N = 158)	125 (79.11%)	33 (20.89%)	

Note—Data are number (%) of target lesions.

**Table 4 cancers-16-00655-t004:** Biopsy outcomes based on initial visualization of each target lesion on digital mammography (DM) or tomosynthesis (DBT).

		DM	DBT	*p* Value, Benign: Cancer
Mass	Benign (N = 89)	44 (72.13%)	45 (69.23%)	0.85
	Cancer (N = 37)	17 (27.87%)	20 (30.77%)	
	Total (N = 126)	61 (48.41%)	65 (51.59%)	
Asymmetry	Benign (N = 150)	77 (80.20%)	73 (74.49%)	0.39
	Cancer (N = 44)	19 (19.80%)	25 (25.51%)	
	Total (N = 194)	96 (49.48%)	98 (50.52%)	
Architectural Distortion	Benign (N = 225)	40 (71.43%)	185 (75.20%)	0.61
	Cancer (N = 77)	16 (28.57%)	61 (24.80%)	
	Total (N = 302)	56 (18.54%)	246 (81.46%)	
Combined	Benign (N = 464)	161 (75.59%)	303 (74.08%)	0.70
	Cancer (N = 158)	52 (24.41%)	106 (25.92%)	
	Total (N = 622)	213 (34.24%)	409 (65.76%)	
*p* value Total DM: Total DBT	Mass vs. Asymmetry			0.91
	Distortion vs. Mass			**<0.001**
	Distortion vs. Asymmetry			**<0.001**

Note—Data are number (%) of target lesions. Boldface indicates statistical significance (*p* ≤ 0.05).

## Data Availability

The original contributions presented in the study are included in the article. Further inquiries can be directed to the corresponding authors.

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
