# Peer review of "Suspicious Ultrasound-Occult Non-Calcified Mammographic Masses, Asymmetries, and Architectural Distortions Are Moderate Probability for Malignancy"

_cancers, 2024, doi:10.3390/cancers16030655_

Round 1

Reviewer 1 Report

Comments and Suggestions for Authors

Your article “Suspicious Ultrasound-occult Non-calcified Mammographic Masses, Asymmetries, and Architectural Distortions are Moderate Probability for Malignancy” is an important study which shows that possible ultrasound correlates demonstrate the increased probability of breast cancer. This is particularly important in patients of increasing age.

Author Response

Your article “Suspicious Ultrasound-occult Non-calcified Mammographic Masses, Asymmetries, and Architectural Distortions are Moderate Probability for Malignancy” is an important study which shows that possible ultrasound correlates demonstrate the increased probability of breast cancer. This is particularly important in patients of increasing age.

Thank you for your time and feedback,

-The Authors.

Reviewer 2 Report

Comments and Suggestions for Authors

What is the main question addressed in this research? The abstract needs modification. Section 2 requires research gap analysis and limitations. Figure 1 needs more on the discussion side. In table 3 why p-Value are in the higher side? this may requires  Friedman test or hypothesis testing. Table 2 requires reasons of the biopsy problems along with explanations. Discussion part has to be enhanced. The  Figure 3 Flowchart demonstrating outcomes for suspicious ultrasound-occult architectural 224 distortions. What is the wishes of the author about the above flow chart. The conclusion needs modifications.

Comments on the Quality of English Language

Nil

Author Response

What is the main question addressed in this research?

              The principal question we addressed in this research is what is the likelihood of malignancy for suspicious, ultrasound-occult, mammographic masses, asymmetries, and architectural distortions (AD).

The abstract needs modification.

              *See below.

Section 2 requires research gap analysis and limitations.

              Section 2 is our materials and methods section. We are unfamiliar with the term gap analysis, so we have done the following research: One of our authors is a biostatistician, and he and his department chair have affirmed that gap analysis is not a statistical tool relevant to our study. An in-depth google search indicates that gap analysis in business and management literature is a process by which businesses can analyze and begin to minimize the “gap” between their current and desired performance. Overall, the goal of our study was not to identify methods to improve radiologist or imaging performance but rather to analyze specific imaging findings and their likelihood of malignancy. Also, we are a bit confused by the “limitations” portion of this comment. Our limitations section is located at the end of the discussion, and we are certainly open to modifying or enhancing it if the reviewer feels that important limitations have been excluded or that unimportant limitations have been overemphasized. We are open to further guidance here.

Figure 1 needs more on the discussion side.

              We have included text under section 3.1 to reflect this change. Thank you.

In table 3 why p-Value are in the higher side? this may require Friedman test or hypothesis testing.

              Table 3 evaluates the likelihood of malignancy for each mammographic finding based on whether it was first visualized on a screening or diagnostic examination. Inherently, the pretest probability for breast cancer at diagnostic mammography is much higher than at screening mammography given that diagnostic mammography patients are typically symptomatic or have a history of important imaging findings. We performed the analysis in table 3 to evaluate if our data is concordant with this trend, and we were surprised to learn that our data is discordant, thus the reason why the p-values are high. We have discussed this finding in the fourth paragraph of the discussion, and we included potential reasons for the statistical insignificance.

Table 2 requires reasons of the biopsy problems along with explanations.

              We are unsure of the exact meaning of this comment. There were no specific problems with the biopsy procedures in our study, and all were successful. Alternatively, perhaps the reviewer is referring to the lesion characteristics that resulted in biopsied lesions being classified as suspicious and warranting biopsy. If so, these characteristics are detailed for each lesion type in the third paragraph of section 2. We are open to further guidance here if the reviewer can provide more specifics. Thank you.

Discussion part has to be enhanced.

              *See below.

The Figure 3 Flowchart demonstrating outcomes for suspicious ultrasound-occult architectural 224 distortions. What is the wishes of the author about the above flow chart.

              Figure 3 reveals the outcomes of all suspicious ADs included in our study. We believe that a big reason why our manuscript should be published is that it represents the largest cohort of ultrasound-occult ADs, which is important because tomosynthesis, which identifies more ADs than digital mammography, continues to rise in popularity throughout the world. The outcome of each of these cases will be of high interest to breast radiologists and breast cancer clinicians because this lesion type is increasingly encountered in clinical practice.

The conclusion needs modifications.
              *See below.

*We appreciate your opinions and comments above and acknowledge that the version of our submitted manuscript is and was not yet ready for publication. However, we politely request a bit more guidance regarding details for how to modify and enhance the abstract, discussion, and conclusion. Specifically, does the reviewer believe we have omitted vital concepts in the discussion or have we overstated/overemphasized our conclusions? Thank you. We are open to further guidance here.

Thank you for your time and feedback,

-The Authors.

Reviewer 3 Report

Comments and Suggestions for Authors

Dear Authors,

Congratulations on your hard work. Here are my suggestions on your manuscript:

- Provide a brief summary of key patient demographics and lesion characteristics.

- Consider discussing any potential limitations or challenges encountered during the retrospective review.

- in the Discussion section a few parragraphs could be integrated to strengthen the discussion, particularly in relation to accuracy and postoperative histopathology in breast cancer patients after neoadjuvant chemotherapy (e.g. Rashmi S, Kamala S, Murthy SS, Kotha S, Rao YS, Chaudhary KV. Predicting the molecular subtype of breast cancer based on mammography and ultrasound findings. Indian J Radiol Imaging. 2018 Jul-Sep;28(3):354-361. doi: 10.4103/ijri.IJRI_78_18. PMID: 30319215; PMCID: PMC6176670.; Schmidt G, Findeklee S, del Sol Martinez G, Georgescu M-T, Gerlinger C, Nemat S, Klamminger GG, Nigdelis MP, Solomayer E-F, Hamoud BH. Accuracy of Breast Ultrasonography and Mammography in Comparison with Postoperative Histopathology in Breast Cancer Patients after Neoadjuvant Chemotherapy. Diagnostics. 2023 https://doi.org/10.3390/diagnostics13172811; Tasoulis MK, Lee HB, Yang W, et al. Accuracy of Post-Neoadjuvant Chemotherapy Image-Guided Breast Biopsy to Predict Residual Cancer. JAMA Surg. 2020;155(12):e204103. doi:10.1001/jamasurg.2020.4103 13(17):2811. 

Author Response

Dear Authors,

Congratulations on your hard work. Here are my suggestions on your manuscript:

Thank you.

- Provide a brief summary of key patient demographics and lesion characteristics.

We have included this summary within 3.1 study population.

- Consider discussing any potential limitations or challenges encountered during the retrospective review.

The last paragraph of our discussion section contains many limitations of our study, and we believe we have discussed the most important of these (related to study design and our topic, suspicious ultrasound-occult mammographic findings). Every study has more limitations and challenges than are explicitly acknowledged in the limitations section, but we politely request to keep the limitations section as currently written unless the reviewer feels strongly that one or more specific limitations have yet to be acknowledged and that including them would increase the value of the manuscript. We are open to further guidance here.

- in the Discussion section a few paragraphs could be integrated to strengthen the discussion, particularly in relation to accuracy and postoperative histopathology in breast cancer patients after neoadjuvant chemotherapy (e.g. Rashmi S, Kamala S, Murthy SS, Kotha S, Rao YS, Chaudhary KV. Predicting the molecular subtype of breast cancer based on mammography and ultrasound findings. Indian J Radiol Imaging. 2018 Jul-Sep;28(3):354-361. doi: 10.4103/ijri.IJRI_78_18. PMID: 30319215; PMCID: PMC6176670.; Schmidt G, Findeklee S, del Sol Martinez G, Georgescu M-T, Gerlinger C, Nemat S, Klamminger GG, Nigdelis MP, Solomayer E-F, Hamoud BH. Accuracy of Breast Ultrasonography and Mammography in Comparison with Postoperative Histopathology in Breast Cancer Patients after Neoadjuvant Chemotherapy. Diagnostics. 2023 https://doi.org/10.3390/diagnostics13172811; Tasoulis MK, Lee HB, Yang W, et al. Accuracy of Post-Neoadjuvant Chemotherapy Image-Guided Breast Biopsy to Predict Residual Cancer. JAMA Surg. 2020;155(12):e204103. doi:10.1001/jamasurg.2020.4103 13(17):2811.

These references are very good studies. However, we are a bit confused as to how to integrate the findings and conclusions from these studies into our work. The Rashmi et al paper deals with predicting breast cancer molecular subtype based on MG and US features, and the main mammographic feature of interest in this work is the presence of calcifications. Our study differs in that we are evaluating non-calcified mammographic asymmetries, distortions, and masses without correlating with molecular subtypes. The latter two references deal specifically with the ability of imaging and percutaneous biopsy to predict the extent of residual breast malignancy following neoadjuvant therapy. However, our study does not pertain to these topics. We are very willing to improve our discussion but ask for further guidance before making these changes so we can incorporate this suggestion in a meaningful way.

Thank you for your time and feedback,

-The Authors.

Round 2

Reviewer 2 Report

Comments and Suggestions for Authors

All the corrections are included in the paper. Hence, the paper is not required for further review.

Comments on the Quality of English Language

Nil